# LEARNING TO DISCOVER REGULATORY ELEMENTS FOR GENE EXPRESSION PREDICTION

**Xingyu Su**[1*]**, Haiyang Yu**[1*]**, Degui Zhi**[2] **& Shuiwang Ji**[1†]
[1]Texas A&M University, [2]The University of Texas Health Science Center at Houston
`{xingyu.su,haiyang,sji}@tamu.edu,{degui.zhi}@uth.tmc.edu`

## ABSTRACT

We consider the problem of predicting gene expressions from DNA sequences. A key challenge of this task is to find the regulatory elements that control gene expressions. Here, we introduce Seq2Exp, a **Seq**uence to **Exp**ression network explicitly designed to discover and extract regulatory elements that drive target gene expression, enhancing the accuracy of the gene expression prediction. Our approach captures the causal relationship between epigenomic signals, DNA sequences and their associated regulatory elements. Specifically, we propose to decompose the epigenomic signals and the DNA sequence conditioned on the causal active regulatory elements, and apply an information bottleneck with the Beta distribution to combine their effects while filtering out non-causal components. Our experiments demonstrate that Seq2Exp outperforms existing baselines in gene expression prediction tasks and discovers influential regions compared to commonly used statistical methods for peak detection such as MACS3. The source code is released as part of the AIRS library (https://github.com/divelab/AIRS/).

## 1 INTRODUCTION

Gene expression serves as a fundamental process that dictates cellular function and variability, providing insights into the mechanisms underlying development (Pratapa et al., 2020), disease (Cookson et al., 2009; Emilsson et al., 2008), and responses to external factors (Schubert et al., 2018). Despite the critical importance of gene expression, predicting it from genomic sequences remains a challenging task due to the complexity and variability of regulatory elements involved. Recent advances in deep learning techniques (Avsec et al., 2021; Gu & Dao, 2023; Nguyen et al., 2024; Badia-i Mompel et al., 2023) have shown remarkable capabilities and performance in modeling long sequential data like language and DNA sequence. By capturing intricate dependencies within genomic data, these techniques provide a deeper understanding of how regulatory elements contribute to transcription (Aristizabal et al., 2020).

To predict gene expression, DNA language models are usually applied to encode long DNA sequences with a subsequent predictor to estimate the gene expression values (Avsec et al., 2021; Nguyen et al., 2024; Gu & Dao, 2023; Schiff et al., 2024). However, those language models are typically designed to encode DNA sequences alone, overlooking the specific environments like different cell types, which leads to suboptimal performance. Instead of predicting the gene expression only using DNA sequence, which is invariant across cell types, a more biological relevant formulation is to predict gene expression levels using both DNA sequence and epigenomic signals. For example, GraphReg (Karbalayghareh et al., 2022) uses epigenomic signals as input data to predict gene expression values. However, it does not integrate DNA sequences and epigenomic signals in a unified manner to improve gene expression prediction. EPInformer (Lin et al., 2024) uses statistical methods to identify the epigenomic signal peaks, and focuses on regulatory elements identified by those peaks. Although obtaining better results, EPInformer still neglects the complex relationship between DNA sequences, epigenomic signals and regulatory elements, which is essential for improving prediction accuracy.

---

[*]Equal contribution.
[†]Corresponding author.

The task of predicting gene expression levels given the DNA sequences and epigenomic signals presents several challenges. First, epigenomic signals can be measured by a variety of experimental techniques, including ChIP-seq, DNase-seq, Hi-C, each with their own biases and limitations (Consortium et al., 2012; Bernstein et al., 2010; Moore et al., 2020). Additionally, the regulatory elements influencing target gene expression are often sparse and may involve long-range interactions, making them challenging to identify and integrate into predictive models. These complexities highlight the need for models that can effectively discover the actively interacted regulatory elements with the target gene on long DNA sequences.

In response to these challenges, we propose Seq2Exp (**Seq**uence to **Exp**ression), a novel framework designed to improve gene expression prediction by selectively extracting relevant sub-sequences from both DNA sequences and epigenomic signals. Since DNA sequences and epigenomic signals capture different aspects of biological information, their integration offers deeper insights. For example, Hi-C/HiChIP data reveals the physical interaction frequency between distal DNA regions, and DNase-seq reflects the functional activity of regulatory elements. Effectively incorporating these signals along with DNA sequences can be highly beneficial for addressing the above challenges for gene expression prediction task. Specifically, in this work, we suggest the causal relationship between genomic data and gene expression to guide the learning process as depicted in Figure 1. Inspired by the causal relationship, we decompose the mask learning process into two components: one based on DNA sequences and the other on epigenomic signals. The proposed Seq2Exp first employs a generator module to learn a token-level mask based on both DNA sequences and epigenomic signals, to extract DNA sub-sequences. Then, the predictor module is applied on these extracted sub-sequences to predict gene expression. With information bottleneck, Seq2Exp can effectively filter out non-causal parts by constraining the mask size, ensuring that only the most relevant regions are extracted. Overall, the incorporation of the DNA sequences and epigenomic signals systematically discovers regions that are likely to influence gene expression.

We summarize our contributions here:

- We propose a framework articulating the causal relationship between epigenomic signals, DNA sequences, target gene expression and related regulatory elements.

- Based on the causal relationships, our framework is proposed to combine the mask probability distribution from DNA sequences and epigenomic signals, and filtering out non-causal region via information bottleneck.

- The proposed Seq2Exp achieves SOTA performances compared to previous gene expression prediction baselines, and demonstrates the extracted regulatory elements serve as a better sub-sequences compared to statistical peaks calling methods of epigenomic signals such as MACS3.

## 2 RELATED WORKS AND PRELIMINARY

### 2.1 TASK DESCRIPTION

Let $X_{\text{seq}} = [x_1, \cdots, x_L]$ denote the DNA sequence with length $L$, where each token $x_i \in \mathbb{R}^{4 \times 1}$ is a one-hot vector representing a nucleotide from the set {A, C, G, T}. For this DNA sequence, the corresponding epigenomic signals are denoted as $X_{\text{sig}} = [s_1, \cdots, s_L]$, where $s_i \in \mathbb{R}^{d \times 1}$ represents $d$ different signals. By using both the DNA sequence and epigenomic signals, the task aims to predict the target gene expression denoted as $Y \in \mathbb{R}$. To achieve this target, we propose our framework to extract the active regulatory elements by learning a token-level binary mask $M = [m_1, \cdots, m_L]$, where $m_i \in \{0, 1\}$ or a soft mask $M$ where $m_i \in [0, 1]$.

Specifically, in our implementation, each example contains one target gene. We first identify the transcription start site (TSS) of the target gene, then select input sequences $X_{\text{seq}}$ and $X_{\text{sig}}$ consist of $L = 200,000$ base pairs, centered on the TSS. Then, the entire sequences provide sufficient contextual information for accurate prediction of the target gene expression value $Y$.

### 2.2 RELATED WORKS

**DNA language model** has been proposed recently to apply language machine learning models to long DNA sequences (Nguyen et al., 2024; Gu & Dao, 2023; Schiff et al., 2024) and solve vari-

ous downstream tasks. Two notable methods in this area are HyenaDNA (Nguyen et al., 2024) and Caduceus (Schiff et al., 2024). HyenaDNA utilizes the Hyena operator (Poli et al., 2023) to process long DNA sequences. Caduceus introduces bidirectional Mamba (Gu & Dao, 2023) for DNA sequences, providing linear complexity for long sequence modeling while also considering the reverse complement of the DNA sequences. These methods offer a powerful approach for modeling long sequence data, such as DNA, and can be fine-tuned for tasks like gene expression prediction. However, they usually only considers DNA sequences as input, and do not explicitly consider the additional epigenomic signals during the prediction. Since these signals often carry meaningful information, such as physical interaction frequency and functional activity, incorporating them into the model could further enhance its performance on the gene expression prediction task.

**Gene expression prediction** is one of the fundamental tasks in bioinformatics (Segal et al., 2002). Numerous studies (Agarwal & Shendure, 2020; Karbalayghareh et al., 2022; Avsec et al., 2021; Lin et al., 2024) have attempted to predict gene expression values directly from DNA sequences. Enformer (Avsec et al., 2021), for instance, tries to only encode DNA sequences as input and employs convolutional and transformer blocks to predict 5,313 human genomic tracks and 1,643 mouse tracks. In contrast, GraphReg (Karbalayghareh et al., 2022), incorporates a graph attention network to account for Hi-C/HiChIP interactions between DNA sub-sequences, improving gene expression predictions by considering physical interaction frequencies. However, both methods either rely on epigenomic signals or DNA sequences as input data, without integrating both data types. Recently, EPInformer (Lin et al., 2024) has advanced this approach by integrating both DNA sequences and epigenomic signals for gene expression prediction. EPInformer first identifies enhancer regions from the DNA sequences based on DNase-seq signals, treating epigenomic signals as enhancer features, and then use promoter-enhancer interactions for gene expression prediction. Despite this progress, EPInformer selects enhancer regions solely based on epigenomic signal peaks, overlooking the complex relationships between DNA sequences, epigenomic signals, and predicted gene expression values. This highlights the need for machine learning methods capable of learning to extract relevant regions in a more comprehensive manner.

## 2.3 BACKGROUND OF INFORMATION BOTTLENECK

To effectively extract active regulatory elements from DNA sequences, it is important to understand the concept of the information bottleneck. The information bottleneck method is a widely used technique in machine learning on tasks for images (Alemi et al., 2016; Chen et al., 2018), language data (Belinkov et al., 2020; Lei et al., 2016; Paranjape et al., 2020; Bastings et al., 2019; Jain et al., 2020) or graph data (Wu et al., 2020; Miao et al., 2022). Its goal is to maximize the mutual information between compressed representations $Z$ and the target variable $Y$, expressed as $I(Z;Y)$, while controlling the information extracted from the input $X$. Note that in the proposed method, $Y$ represents the target gene expression. A straightforward approach would be to set $Z = X$, but this retains the full complexity of $X$, which makes the optimization process challenging, especially with the long and noisy nature of DNA sequences.

To address this, researchers impose a constraint on the information transferred from $X$ to $Z$, ensuring that $I(X;Z) \leq I_c$, where $I_c$ is an information constraint that allows us to capture only the most critical compressed representations. The information bottleneck objective becomes maximizing:

$$L = I(Z;Y) - \beta I(X;Z), \tag{1}$$

where $\beta$ is a hyperparameter that balances the trade-off between compression and relevance. However, directly optimizing this objective is challenging. To overcome this, Chen et al. (2018) proposes to maximize a lower bound approximation, which leads to minimizing the following expression:

$$L \approx \frac{1}{N} \sum_{i=1}^{N} \mathbb{E}_{p_\theta(Z|x_i)}[-\log q_\phi(y_i|Z)] + \beta KL[p_\theta(Z|x_i), r(Z)], \tag{2}$$

where $p_\theta(Z|x_i)$ is a parametric approximation of $Z$, $q_\phi(y_i|Z)$ is a variational approximation of the true distribution $p(y_i|Z)$, and $r(Z)$ approximates the marginal distribution $p(Z)$.

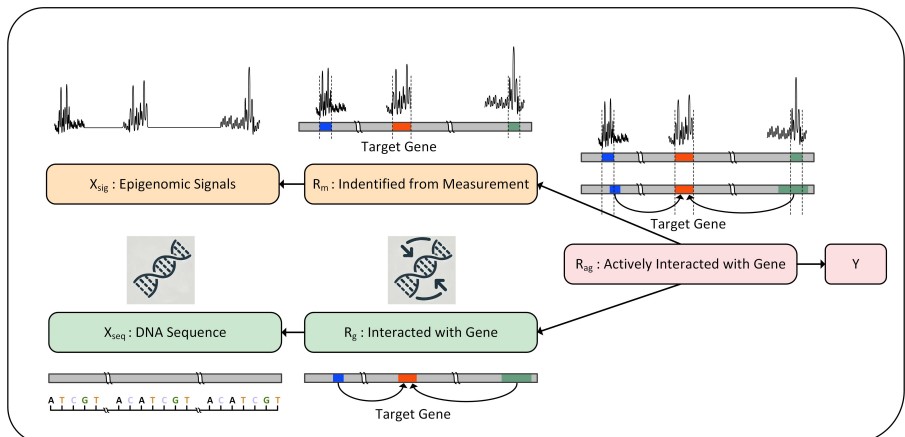

Figure 1: Causal relationships between epigenomic signals, sequence, gene expression $Y$ and related regulatory elements.

## 3 PROPOSED METHODS

In this section, we present our framework Seq2Exp. We first present our motivation for predicting gene expression with learnable extraction of effective regulatory elements. We illustrate the causal relationship among regulatory elements, epigenomic signals and DNA sequences as shown in Figure 1. Motivated by this structural causal model (SCM) (Pearl, 2009; Pearl et al., 2000; Wu et al., 2022), our framework provides a learnable approach to effectively extract effective regulatory elements, considering both DNA sequences and epigenomic signals, through an information bottleneck mechanism.

### 3.1 CAUSAL RELATIONSHIP AMONG REGULATORY ELEMENTS, DNA SEQUENCE AND EPIGENOMIC SIGNALS

The interactions between target gene and regulatory elements are complex, particularly when multiple potential regulatory elements are involved. Meanwhile, long sequences and distal interactions require a large search region, further complicating the discovery of effective regulatory elements that influence target gene expression. In this study, we take use of epigenomic signals $X_{\text{sig}}$ from laboratory experiments as well as the DNA sequence $X_{\text{seq}}$ for target gene expression $Y$, and formulate their relationships with the proposed three categories of regulatory elements.

- $R_{\text{g}}$: Regulatory elements that have the potential to interact with target gene. However, they might not influence target gene expression if they are inactive in a specific cell type or are distant from the target gene.

- $R_{\text{m}}$: Regulatory elements discovered from measurement. Typically, the region with strong measured epigenomic signals, such as peaks in DNase-seq, are more likely to influence the gene expression. However, there are usually multiple genes within a sequence and the association of $R_{\text{m}}$ with target gene remains unknown.

- $R_{\text{ag}}$: Regulatory elements actively interacted with target gene. It is identified as the causal component for the final target gene expression $Y$.

The causal relationship between these variables is depicted in Figure 1. Note that each variable corresponds to a distribution and link represents a causal connection. The flow of this SCM illustrates the perspective of data generation.

- $X_{\text{seq}} \longleftarrow R_{\text{g}}$. The DNA sequence consists of $R_{\text{g}}$ and other non-causal parts.
- $R_{\text{ag}} \longrightarrow Y$. The causal part $R_{\text{ag}}$ directly influences the final gene expression. For example, an active enhancer interacting with a gene can significantly impact its expression.
- $R_{\text{g}} \longleftarrow R_{\text{ag}} \longrightarrow R_{\text{m}}$. The key causal component $R_{\text{ag}}$ is shared by both $R_{\text{g}}$ and $R_{\text{m}}$. It can be detected through epigenomic signals in laboratory experiments and also participates in interactions with the target gene.

- $R_{\mathrm{m}} \longrightarrow X_{\mathrm{sig}}$. $X_{\mathrm{sig}}$ usually contains strong observable signals, such as peaks in DNase-seq, whereas regions without such signals often provide limited useful information.

## 3.2 TASK OBJECTIVE

Based on information bottleneck, Equation 2 describes how to learn compressed representations $Z$ rather than selecting specific sub-sequences. To directly select regulatory elements, we define the latent representations as $Z = M \odot X$, where $M$ is a binary variable controlling the selection of each DNA base or a soft mask $M$ indicating the importance of each DNA base. We assume that each selection is independent given the input sequence $X$, i.e., $p(M|X) = \prod_i p(m_i|X)$. Following the method of Paranjape et al. (2020), the objective becomes:

$$L \approx \frac{1}{N} \sum_{i=1}^{N} \mathbb{E}_{p_\theta(m_i|x_i)}[-\log q_\phi(y_i|m_i \odot x_i)] + \beta KL[p_\theta(m_i|x_i), r(m_i)], \tag{3}$$

where the first term is the task-specific loss, such as mean square error in DNA gene expression prediction, and the second term imposes a constraint on the learned mask $m$, aligning it with the predefined distribution $r(m)$ without conditioning on any specific sequence $x$. In our case, we use this second term to enforce sparsity in the learned regulatory elements.

## 3.3 DECOMPOSITION OF SEQUENCES AND SIGNALS

By using information bottleneck shown in Equation 3, our primary focus is on estimating $p_\theta(M|X)$, i.e., learning the mask from the input sequences. Given that the input $X$ consists of both DNA sequences and epigenomic signals, we need to estimate $p_\theta(M|\{X_{seq}, X_{sig}\})$.

**Assumption 1** (Conditional Independence of Sequences and Signals). *We assume that, conditioned on the selection of regulatory elements $M$, the DNA sequences and epigenomic signals are conditional independent to each other, i.e.,*

$$p(X_{sig}, X_{seq}|M) = p(X_{sig}|M)p(X_{seq}|M) \tag{4}$$

Assumption 1 is based on the causal relationships outlined in Section 3.1. The selected sub-sequences of a full given sequence, represented by $M \odot X$, can be viewed as the optimal regulatory elements ($R_{ag}$) for a specific gene in a particular cell type. From a data generation perspective, both the regulatory elements detected through measurements ($R_m$) and those interacting with the gene ($R_g$) originate from the optimal regulatory elements ($R_{ag}$). Therefore, given the optimal regulatory elements, the distributions $p(X_{sig}|M)$ and $p(X_{seq}|M)$ should be independent of each other.

**Proposition 1.** *Based on Assumption 1, the estimation of $p_\theta(M|X)$ can be decomposed into terms involving $X_{seq}$ and $X_{sig}$. Specifically, we have*

$$p_\theta(M|X) \propto p_{\theta_1}(M|X_{seq})p_{\theta_2}(M|X_{sig}), \tag{5}$$

*where $p_{\theta_1}(M|X_{seq})$ and $p_{\theta_2}(M|X_{sig})$ represent the contributions from the DNA sequence and the epigenomic signals, respectively.*

The detailed proof of this decomposition is provided in Appendix A.1. Proposition 1 allows us to factorize the estimation of $p_\theta(M|X)$ into two independent components, corresponding to the DNA sequence $X_{seq}$ and the epigenomic signals $X_{sig}$. As a result, we can independently estimate $p_{\theta_1}(M|X_{seq})$ and $p_{\theta_2}(M|X_{sig})$, which simplifies the overall estimation process. This decomposition is based on the assumption that, conditioned on the selection of regulatory elements $m$, the DNA sequences and epigenomic signals are independent, thus enabling more efficient and targeted modeling of each component.

## 3.4 MASK DISTRIBUTION

With the conditional independence property shown in Proposition 1, the estimation of the mask $M$ can be decomposed into two components: one based on DNA sequences $p_{\theta_1}(M|X_{seq})$ and the other on epigenomic signals $p_{\theta_2}(M|X_{sig})$. We assume that both components follow the Beta distribution, as described in Assumption 2. The sampled values from the Beta distribution represent the probability of selecting specific base pairs from a DNA sequence.

**Assumption 2** (Mask Distribution). *We assume that the soft mask $m_s$ follows the Beta distribution, i.e., $m_s \sim Beta(\alpha, \beta)$.*

Unlike the binary hard mask $M$, the soft mask $m_s$ takes values between 0 and 1, making it more suitable for the Beta distribution. The hard mask $M$ can then be obtained by applying a threshold to the soft mask. For the implementation, we apply both hard mask version and soft mask version.

There are several reasons for choosing the Beta distribution. First, the Beta distribution typically quantifies success rates (DeGroot & Schervish, 2013; Gelman et al., 2013). The input parameters $\alpha$ and $\beta$ represent the weights for selecting and not selecting the base pair, respectively. Therefore, when $\alpha > \beta$, the base pair is more likely to be selected, and vice versa. Second, as both $\alpha$ and $\beta$ increase simultaneously, the selection process will exhibit lower variance, indicating more confidence in the selection. Third, the product of two Beta distributions, when properly normalized, results in another Beta distribution. This ensures that the distributions within the framework remain in the same family, simplifying subsequent mathematical calculations and providing consistent fitting objectives for the models.

Based on these properties of the Beta distribution, we assume that both $p_{\theta_1}(m_s|X_{seq})$ and $p_{\theta_2}(m_s|X_{sig})$ follow Beta distributions, but with different parameters $\alpha$ and $\beta$.

**Proposition 2.** *Given $p_{\theta_1}(m_s|X_{seq}) \sim Beta(\alpha_1, \beta_1)$ and $p_{\theta_2}(m_s|X_{sig}) \sim Beta(\alpha_2, \beta_2)$, the product of these distributions also follows a Beta distribution, with parameters:*

$$p_{\theta_1}(m_s|X_{seq})p_{\theta_2}(m_s|X_{sig}) \sim Beta(\alpha_1 + \alpha_2 - 1, \beta_1 + \beta_2 - 1) \tag{6}$$

The proof of Proposition 2 is provided in Appendix A.2. When combining the probability distributions learned from the DNA sequence and signals, Proposition 2 ensures that the resulting distribution remains within the same family. And the final mask $m_s$ is then obtained through the combined Beta distribution. Specifically, deep learning models are applied in our framework to learn these two distributions by predicting the parameters $\alpha$ and $\beta$.

### 3.5 SPARSE OBJECTIVE

In this part, we focus on the mask prior distribution $r(m)$. From the objective in Equation 3, the KL divergence between $p_\theta(m_i|x_i)$ and $r(m_i)$ needs to be computed. To simplify this calculation, we assume the prior distribution of the soft mask $r(m_s)$ follows the Beta distribution as well. Therefore, we have $r(m_s) \sim \text{Beta}(\alpha_3, \beta_3)$, where $\alpha_3$ and $\beta_3$ are related to the sparsity of mask.

The expectation of the Beta distribution is

$$\mathbb{E}[m_s] = \mu = \frac{\alpha_3}{\alpha_3 + \beta_3}, \tag{7}$$

where $\mu$ approximately represents the proportion of regulatory elements within the DNA sequences. Therefore, by setting the hyperparameters $\alpha_3$ and $\beta_3$, the sparsity of the mask is taken into consideration, acting as a bottleneck to filter out non-causal parts.

## 4 MODEL DESIGNS

### 4.1 ARCHITECTURE

As shown in Figure 2, our proposed model generate the mask distribution $p_\theta(M|X)$ from the DNA sequences and epigenomic signals $X = \{X_{seq}, X_{sig}\}$, and an predictor, $q_\phi(Y|M \odot X)$, provides gene expression values from the masked sequences $Z = M \odot X$. Those two modules will be trained together.

**Generator.** As outlined in Section 3.4, we aim to generate a mask $M$ to identify the critical regions within the DNA sequences. To achieve this, we first learn a soft mask $m_s$, which is a probabilistic representation of each base pair's relevance, where $m_s \in [0, 1]$. The soft mask is modeled using the Beta distribution, whose parameters—$\alpha_1$, $\alpha_2$, $\beta_1$, and $\beta_2$—are estimated from the combination of DNA sequences and epigenomic signals, as detailed in Proposition 2.

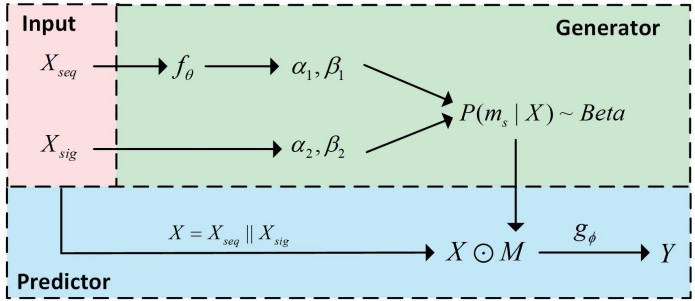

Figure 2: Pipeline of proposed architectures. The input data contains the DNA sequence $X_{\text{seq}}$ and epigenomic signals $X_{\text{sig}}$. A deep learning model $f_\theta$ is then applied to $X_{\text{seq}}$ to learn the corresponding parameters for the Beta distribution $\alpha_1, \beta_1$, while $\alpha_2, \beta_2$ are obtained from $X_{\text{sig}}$ in a non-parameterized manner. By combining these two beta distributions, $p(m_s|X)$ is obtained and used to generate the mask for actively interacted regulatory elements. The selected elements are then fed into the predictor model $g_\phi$ to provide the final target gene expression.

For the parameters derived from the DNA sequences, the neural network $f_\theta$ is used to predict $\alpha_1$ and $\beta_1$. Specifically, we have

$$\alpha_1, \beta_1 = f_\theta(X_{seq}), \tag{8}$$

where the network $f_\theta$ outputs the $L$-dimensional parameters $\alpha_1$ and $\beta_1$, with $L$ being the length of the input DNA sequence. Each position in the sequence is associated with a pair of values $\alpha_1$ and $\beta_1$, which parameterize the Beta distribution for that base pair.

For the parameters related to epigenomic signals, we use the intuition that higher signal values increase the likelihood of selecting the corresponding base pair. To capture this, we directly use the epigenomic signal values as the parameter $\alpha_2$, which influences the selection weight for each base pair. The parameter $\beta_2$, representing a selection threshold, is set as a fixed constant. Specifically, we define

$$\alpha_2 = X_{sig}; \beta_2 = C_\beta. \tag{9}$$

By the above modeling procedure, we simplify the modeling process, making the learning of $\alpha_2$ and $\beta_2$ non-parametric while maintaining the influence of signal strength without introducing additional learnable parameters.

After estimating the parameters, based on Proposition 2, the soft mask $m_s$ is sampled from the combined Beta distribution, $p_{\theta_1}(m_s|X_{seq})p(m_s|X_{sig}) \sim \text{Beta}(\alpha_1 + \alpha_2 - 1, \beta_1 + \beta_2 - 1)$, which represents the probability of selecting each base pair in the sequence. This probabilistic formulation allows us to model the selection process effectively.

Finally, for the hard mask version, a threshold is applied to the soft mask to generate the hard mask, $M = \mathbb{I}(m_s \geq C_m)$, where $C_m$ is the threshold (e.g., 0.5). The hard mask $M$ provides a binary decision for selecting or ignoring specific base pairs. Through this approach, we model the mask generation process by leveraging both DNA sequences and epigenomic signals, combining parametric and non-parametric methods for more efficient region selection.

**Predictor.** After obtaining the mask $M$, we apply it to the input sequences to extract the relevant sub-sequences, represented as $M \odot X$. The extracted sub-sequences are then fed into a secondary neural network, denoted by $g_\phi$, to estimate the probability distribution of the target gene expression $Y$. The conditional distribution is expressed as $q_\phi(Y|M \odot X)$, where $\phi$ represents the parameters of the network, and $M \odot X$ refers to the masked input sequences.

To incorporate epigenomic signals alongside the DNA sequences, the input $X$ is formed by concatenating the one-hot encoded DNA sequence embeddings with the epigenomic signal values. This combined input allows the model to leverage both DNA sequence information and epigenomic signals, enhancing the model's predictive capability during the estimation process.

## 4.2 OPTIMIZATION

To optimize the loss function introduced in Equation 3, it is essential that every step remains differentiable to allow for gradient-based optimization during training. After obtaining the parameters

of the Beta distribution through the neural network $p_\theta$, we generate the soft mask $m_s$ by sampling from this distribution. To maintain differentiability, we treat the Beta distribution as a special case of the Dirichlet distribution (Figurnov et al., 2018; Bishop, 2006). Using the reparameterization trick, we achieve differentiable sampling from the Dirichlet distribution by first sampling from the Gamma distribution and then normalizing the results (Figurnov et al., 2018). This method ensures that the sampling process remains differentiable with respect to the parameters $\alpha$ and $\beta$, allowing for efficient backpropagation and optimization.

During inference, instead of sampling from the Beta distribution, we directly use the *expected value* of the Beta distribution as the soft mask $m_s$ for each base pair. The expected value of a Beta distribution with parameters $\alpha$ and $\beta$ is given by $\mathbb{E}[m_s] = \frac{\alpha}{\alpha+\beta}$, which provides a deterministic and efficient way to generate the soft mask without introducing randomness during inference, thus stabilizing the model's predictions.

For the soft mask version, we multiply the soft mask value. And for the hard mask version, when the soft mask $m_s$ is obtained, we need to convert it into a hard binary mask $M$ to make final selections for each base pair. To retain differentiability in this process, we apply the *straight-through estimator (STE)* commonly used in Gumbel-Softmax (Jang et al., 2016). The STE allows us to make the forward pass non-differentiable by applying a hard threshold (e.g., setting values greater than 0.5 to 1 and others to 0), while during the backward pass, the gradient is propagated through the soft mask as if it were continuous. This approach ensures that the model can learn effectively while using discrete decisions during the forward pass, preserving differentiability in the overall optimization process.

## 5 EXPERIMENTS

### 5.1 SETTINGS

#### 5.1.1 DATASETS AND INPUT FEATURES

In this study, we aim to predict gene expression by modeling CAGE values, as it serves as key indicators of gene expression levels. Since gene expression varies across different cell types, we focus on two well-studied cell types: K562 and GM12878, both commonly used in biological research. The CAGE data are sourced from the ENCODE project (Consortium et al., 2012), and we follow the methodology of Lin et al. (2024) to predict gene expression values for 18,377 protein-coding genes.

For the input data, we utilize the HG38 human reference genome to provide the reference DNA sequences. Additionally, the model incorporates several types of epigenomic signals:

- **DNase-seq** data is used to capture chromatin accessibility by identifying regions of the genome that are open and accessible to transcription factors and other regulatory proteins. The signals are extracted from bigWig files, providing genome-wide distributions of chromatin accessibility.
- **H3K27ac** ChIP-seq data is used to detect histone modifications, specifically the acetylation of lysine 27 on histone H3, which is often associated with active enhancers and promoters. This data is also extracted from bigWig files to provide genome-wide information on histone modification patterns.
- **Hi-C** data is processed to calculate the contact frequencies between each base pair and the target transcription start site (TSS), following the ABC pipeline method as described by Fulco et al. (2019).

Furthermore, we incorporate additional features such as mRNA half-life and promoter activity from previous studies (Lin et al., 2024), which improve the model's prediction accuracy on gene expression levels. The detailed information about these signals can be found in Appendix A.3.

A detailed description of data acquisition, preprocessing, and extraction, including downloading, filtering, and alignment, is provided in Appendix A.3.

#### 5.1.2 BASELINES

To benchmark our model's performance, we compare it against several well-established baselines in gene expression prediction:

- **Enformer** (Avsec et al., 2021): A widely used deep learning model for gene expression prediction, designed to capture long-range interactions across DNA sequences. Enformer employs the CNN and transformer architecture to model the DNA sequence to get the gene expression.
- **HyenaDNA** (Nguyen et al., 2024): A cutting-edge method for modeling long DNA sequences, building on the Hyena (Poli et al., 2023) operator, which introduces a novel way to handle long-range dependencies efficiently. HyenaDNA is designed to maintain high accuracy while significantly reducing computational complexity compared to traditional transformer-based models.
- **Mamba** (Gu & Dao, 2023): A long-sequence modeling approach based on the state space model (SSM), offering linear computational complexity. Mamba is specifically tailored for efficiently handling long sequences, making it highly scalable while retaining strong predictive performance.
- **Caduceus** (Schiff et al., 2024): The state-of-the-art model for long genomic sequence modeling, built upon the Mamba architecture. Caduceus is optimized for learning rich representations of genomic sequences. In our study, we utilize Caduceus-Ph. A classification layer is appended to evaluate its performance on our specific task.
- **EPInformer** (Lin et al., 2024): A recently developed model extends the Activity-By-Contact (ABC) model (Fulco et al., 2019) for gene expression prediction. EPInformer utilizes DNase-seq peak data to define potential regulatory regions and applies an attention mechanism to aggregate enhancer signals. By leveraging both epigenomic and spatial information, EPInformer effectively models the enhancer information for gene expression prediction.

Note that the size of the field of view of Enformer, HyenaDNA, Mamba, Caduceus are long-range DNA sequence, while the EPInformer is the extracted potential enhancer candidates based on DNase-seq measurement following ABC model [5]. Moreover, our proposed Seq2Exp also has the view of long-range DNA sequence, since the generator will take the long-range DNA sequence to extract the relevant sub-sequences for the prediction.

### 5.1.3 Evaluation Metrics

We employ the following evaluation metrics to assess the performance of our model and baselines on the gene expression regression task: Mean Squared Error (MSE) measures the average squared difference between the predicted and target gene expression values, capturing large deviations strongly. Mean Absolute Error (MAE) evaluates the absolute differences between predicted and actual values, providing a more direct measure of average prediction error. Pearson Correlation is used to assess the linear correlation between the predicted and actual gene expression values, reflecting how well the model captures the relative ordering of gene expression. While MSE and MAE focus on the absolute errors in predictions, Pearson Correlation assess the model's ability to capture relative ranking and overall trends in the data.

### 5.1.4 Implementation Details

We evaluate model performance using a cross-chromosome validation strategy. The model is trained on all chromosomes except those designated for validation and testing. Specifically, chromosomes 3 and 21 are used as the validation set, and chromosomes 22 and X are reserved for the test set. The inclusion of chromosome X is particularly challenging due to its unique biological characteristics compared to autosomes, thus providing a more stringent test of the model's robustness.

Both generator $p_\theta$ and predictor $q_\phi$ are based on Caduceus architecture (Schiff et al., 2024), an advanced long-sequence model considering the bi-direction and RC-equivariance for DNA. Specifically, we train for 50,000 steps on a 4-layer Caduceus architecture from scratch with a hidden dimension of 128, and more hyperparameters can be found in the Appendix A.4

All experiments were conducted on a system equipped with an NVIDIA A100 80GB PCIe GPU.

The input sequences consist of 200,000 base pairs, centered around the promoter regions of the target genes, providing sufficient contextual information for accurate gene expression prediction.

### 5.2 Results of Gene Expression Prediction

Table 1 presents the gene expression results based on CAGE values. Enformer, HyenaDNA, Mamba, and Caduceus are all DNA sequence-based methods, relying solely on DNA sequences without incorporating epigenomic signals. Among these, Caduceus achieves the best performance. We

Table 1: Performance on Gene Expression CAGE Prediction. The top performance over all the methods are highlighted in **bold**. Underline indicates that the best performance over all the baselines.

| | K562 | | | GM12878 | | |
|---|---|---|---|---|---|---|
| | MSE ↓ | MAE ↓ | Pearson ↑ | MSE ↓ | MAE ↓ | Pearson ↑ |
| Enformer | 0.2920 | 0.4056 | 0.7961 | 0.2889 | 0.4185 | 0.8327 |
| HyenaDNA | 0.2265 | 0.3497 | 0.8425 | 0.2217 | 0.3562 | 0.8729 |
| Mamba | 0.2241 | 0.3416 | 0.8412 | 0.2145 | 0.3446 | 0.8788 |
| Caduceus | 0.2197 | 0.3327 | 0.8475 | 0.2124 | 0.3436 | 0.8819 |
| Caduceus w/ signals | 0.1959 | 0.3187 | 0.8630 | 0.1942 | 0.3269 | 0.8928 |
| EPInformer | 0.2140 | 0.3291 | 0.8473 | 0.1975 | 0.3246 | 0.8907 |
| Seq2Exp-hard | 0.1863 | 0.3074 | 0.8682 | 0.1890 | 0.3199 | 0.8916 |
| Seq2Exp-soft | **0.1856** | **0.3054** | **0.8723** | **0.1873** | **0.3137** | **0.8951** |

Table 2: Comparison with MACS3 on Gene Expression CAGE Prediction.

| | K562 | | | | GM12878 | | | |
|---|---|---|---|---|---|---|---|---|
| | MSE ↓ | MAE ↓ | Pearson ↑ | Mask Ratio | MSE ↓ | MAE ↓ | Pearson ↑ | Mask Ratio |
| Seq2Exp-hard | 0.1863 | 0.3074 | 0.8682 | 6.88% | 0.1890 | 0.3199 | 0.8916 | 6.32% |
| Seq2Exp-retrain | 0.1979 | 0.3168 | 0.8623 | 10.00% | 0.1887 | 0.3177 | 0.8941 | 10.00% |
| MACS3 | 0.2195 | 0.3455 | 0.8435 | 13.61% | 0.2340 | 0.3654 | 0.8634 | 15.95% |

further evaluate Caduceus by incorporating epigenomic signals, concatenated with the one-hot DNA sequence embeddings as input features. EPInformer, which explicitly extracts enhancer regions based on DNase-seq measurements, outperforms other baselines. This highlights that selecting key regions based on epigenomic signals yields better results.

Finally, our proposed model, Seq2Exp, achieves the best performance overall. By using the Caduceus sequence model as both the generator and predictor, and incorporating epigenomic signals as additional features to the predictor, Seq2Exp explicitly learns the positions of regulatory elements from both DNA sequences and epigenomic signals, resulting in superior performance. We propose two versions of Seq2Exp. Seq2Exp-hard is to have a binary mask, and Seq2Exp-soft takes use of soft mask values to denote the importance, resulting in an even better performances regarding the CAGE prediction task.

### 5.3 Comparison with peak detection method

Table 2 compares the performance of Seq2Exp with regions identified through peak calling by MACS3 (Zhang et al., 2008) on DNase-seq epigenomic signals. While DNase-seq is a crucial technique for identifying the positions of regulatory elements, statistical peak-calling methods, such as MACS3, can be considered a simple approach for measuring these elements. Our results show that Seq2Exp significantly outperforms MACS3 in terms of predictive performance. Seq2Exp-hard utilizes a hard binary mask. Seq2Exp-retrain builds on a soft mask, and explicitly select the top 10% of base pairs and retrain the predictor model using only the selected ones. Both models outperform MACS3, suggesting the ability of discovering regulatory elements.

## 6 Conclusion

In this work, we introduced Seq2Exp, a framework for gene expression prediction that learns critical regulatory elements from both DNA sequences and epigenomic signals. By generating a binary mask to identify relevant sub-sequences, Seq2Exp reduces input complexity and focuses on key regions for prediction. Our experiments demonstrate its effectiveness in identifying important regulatory elements and improving predictive performances, though current evaluations are limited to two cell types and several epigenomic signals.

For the future direction, expanding the framework to more cell types and integrating diverse epigenomic data will be important for validating its generalizability. Beyond gene expression, applying this approach to other tasks related to regulatory element discovery and sequence analysis presents exciting research opportunities. Developing pretraining models focused on regulatory element extraction could also advance the field, enhancing generalization across genomic tasks.

## ACKNOWLEDGMENTS

Research reported in this publication was supported in part by the National Institute on Aging of the National Institutes of Health under Award Number U01AG070112 and ARPA-H under Award Number 1AY1AX000053. The content is solely the responsibility of the authors and does not necessarily represent the official views of the funding agencies.

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

# A    APPENDIX

## A.1    DERIVATION OF SEQUENCE AND SIGNAL DECOMPOSITION

For the mask distribution $p_\theta(m|X)$, we aim to decompose it. For simplicity, we omit the parameter $\theta$ in the following derivation. By applying Bayes' theorem, we obtain

$$
\begin{aligned}
p(m|X) &= p(m|X_{seq}, X_{sig}) \\
&= \frac{p(X_{seq}, X_{sig}|m)p(m)}{p(X_{seq}, X_{sig})} \\
&\propto p(X_{seq}|m)p(X_{sig}|m)p(m),
\end{aligned}
\tag{10}
$$

where $p(X_{seq}, X_{sig}|m) = p(X_{seq}|m)p(X_{sig}|m)$ is based on Assumption 1, and $p(X_{seq}, X_{sig})$ represents the input data, which is independent of the learning process.

Applying Bayes' theorem again to $p(X_{seq}|m)$ and $p(X_{sig}|m)$, we have

$$
\begin{aligned}
p(m|X) &\propto p(X_{seq}|m)p(X_{sig}|m)p(m) \\
&= \frac{p(m|X_{seq})p(X_{seq})}{p(m)} \frac{p(m|X_{sig})p(X_{sig})}{p(m)}p(m) \\
&\propto \frac{p(m|X_{seq})p(m|X_{sig})}{p(m)},
\end{aligned}
\tag{11}
$$

where we can safely omit $p(X_{seq})$ and $p(X_{sig})$. For the marginal distribution $p(m)$, we make it to be a prior distribution with constant predefined parameters, allowing us to omit it as well. Thus, we derive

$$
p(m|X) \propto p(m|X_{seq})p(m|X_{sig}),
\tag{12}
$$

which corresponds to Proposition 1.

## A.2    BETA DISTRIBUTION PRODUCT

The probability density function for a Beta distribution is given by

$$
\text{Beta}(x; \alpha, \beta) \propto x^{\alpha-1}(1-x)^{\beta-1}.
\tag{13}
$$

Given that both $p(m_s|X_{seq})$ and $p(m_s|X_{sig})$ follow a Beta distribution, we have

$$
\begin{aligned}
p(m_s|X_{seq}) &\propto x^{\alpha_1-1}(1-x)^{\beta_1-1}, \\
p(m_s|X_{sig}) &\propto x^{\alpha_2-1}(1-x)^{\beta_2-1}.
\end{aligned}
\tag{14}
$$

Multiplying these distributions yields

$$
\begin{aligned}
p(m_s|X_{seq})p(m_s|X_{sig}) &\propto x^{\alpha_1+\alpha_2-2}(1-x)^{\beta_1+\beta_2-2} \\
&\sim \text{Beta}(m_s; \alpha_1 + \alpha_2 - 1, \beta_1 + \beta_2 - 1).
\end{aligned}
\tag{15}
$$

Note that the parameters of a Beta distribution must lie within the range $(0, \infty)$, thus we require $\alpha_1 + \alpha_2 > 1$ and $\beta_1 + \beta_2 > 1$ to ensure a valid distribution.

## A.3    DATA PROCESSING

The gene expression is different for different cell types. In this work, we consider the well-studied cell type K562 and GM12878.

**CAGE.** Cap Analysis of Gene Expression (CAGE) is the primary target for prediction in this work. Each gene is assigned a CAGE value to quantify its expression level. CAGE is a high-throughput sequencing technique primarily used to map transcription start sites (TSS) and quantify gene expression. It provides a comprehensive profile of promoter usage and alternative TSS across different genes, quantifying the number of RNA molecules initiating at each TSS, thereby reflecting gene transcriptional activity.

Table 3: Hyperparameter values and their search space (final choices are highlighted in **bold**).

| Hyperparameters | Values |
|---|---|
| # Layers of Generator | 4 |
| # Layers of Predictor | 4 |
| Hidden dimensions | 128 |
| $\alpha_3, \beta_3$ | $[1, 9]$, **[10,90]**, $[10, 190]$, $[10, 10]$, $[10, 1.11]$ |
| # training steps | **50000**, $85000$ |
| Batch size | 8 |
| Learning rate | $1e-3$, **5e-4**, $1e-4$, $5e-5$ |
| Scheduler strategy | CosineLR with Linear Warmup |
| Initial warmup learning rate | 1e-5 |
| Min learning rate | 1e-4 |
| Warmup steps | 5,000 |
| Validation model selection criterion | validation MSE |

In this study, we use CAGE data from the FANTOM5 project (Andersson et al., 2014) (K562: CNhs11250; GM12878: CNhs12333). We follow the procedures outlined in Agarwal & Shendure (2020) and Lin et al. (2024) to derive the target values for each gene.

**DNase-seq.** DNase-seq (DNase I hypersensitive site sequencing) is a technique used to identify regions of open chromatin within the genome. It pinpoints areas that are less compacted by nucleosomes, typically corresponding to promoters, enhancers, and transcription factor binding sites. The value derived from DNase-seq represents the frequency of DNase I cleavage at specific sites, with higher values indicating regions that are more accessible to regulatory elements.

We obtained the DNase-seq data from the ENCODE project (Consortium et al., 2012) (K562: ENCFF414OGC; GM12878: ENCFF960FMM). We directly downloaded the data in bigWig format, as it provides a genome-wide distribution of DNase-seq values.

**H3K27ac.** H3K27ac refers to the acetylation of lysine 27 on histone H3, a post-translational modification associated with active enhancers and promoters. High levels of H3K27ac in a genomic region indicate that it is likely an active enhancer or promoter, playing a significant role in gene expression regulation.

We also obtained H3K27ac data from the ENCODE project (Consortium et al., 2012) (K562: ENCFF465GBD; GM12878: ENCFF798KYP), again in bigWig format, which provides the value distribution across the genome.

**Hi-C.** Hi-C measures the three-dimensional (3D) organization of genomes by capturing physical interactions between different regions of DNA. This technique helps researchers understand how DNA is folded and structured within the nucleus. Hi-C data provides information about genome contacts, but due to technical limitations, it often has low resolution (typically at 5,000 base pairs), meaning we can only observe interactions between two regions of DNA of at least this length.

In this work, we follow previous studies (Fulco et al., 2019), calculating the frequency of contacts between a specific region (TSS) and all other regions, generating a Hi-C frequency distribution across the genome.

The Hi-C data were sourced from the 4D Nucleome project (Dekker et al., 2017) (K562: 4DNFI-TUOMFUQ; GM12878: 4DNFI1UEG1HD).

**mRNA half-life and promoter activity features.** When predicting the CAGE values, following the implementation of Lin et al. (2024), we use the promoter activity feature and mRNA half-life features as supplementary for fair comparison and further improvement. The promoter activity feature is defined as the square root of the product of DNase-seq and H3K27ac signal values. The mRNA features include G/C contents, lengths of functional regions, intron length, and exon junction density within the coding region. Specifically, the features are

- The log-transformed z-score of the 5' UTR (untranslated region) length.
- The log-transformed z-score of the CDS (coding sequence) length.

Table 4: Performance on Gene Expression CAGE Prediction with Standard Deviation for Cell Type K562.

|  | MSE ↓ | MAE ↓ | Pearson ↑ |
|---|---|---|---|
| Enformer | $0.2920 \pm 0.0050$ | $0.4056 \pm 0.0040$ | $0.7961 \pm 0.0019$ |
| HyenaDNA | $0.2265 \pm 0.0013$ | $0.3497 \pm 0.0012$ | $0.8425 \pm 0.0008$ |
| Mamba | $0.2241 \pm 0.0027$ | $0.3416 \pm 0.0026$ | $0.8412 \pm 0.0021$ |
| Caduceus | $0.2197 \pm 0.0038$ | $0.3327 \pm 0.0070$ | $0.8475 \pm 0.0014$ |
| Caduceus w/ signals | $\underline{0.1959 \pm 0.0036}$ | $\underline{0.3187 \pm 0.0036}$ | $\underline{0.8630 \pm 0.0008}$ |
| EPInformer | $0.2140 \pm 0.0042$ | $0.3291 \pm 0.0031$ | $0.8473 \pm 0.0017$ |
| MACS3 | $0.2195 \pm 0.0023$ | $0.3455 \pm 0.0018$ | $0.8435 \pm 0.0013$ |
| Seq2Exp-hard | $0.1863 \pm 0.0051$ | $0.3074 \pm 0.0036$ | $0.8682 \pm 0.0045$ |
| Seq2Exp-soft | $\mathbf{0.1856 \pm 0.0032}$ | $\mathbf{0.3054 \pm 0.0024}$ | $\mathbf{0.8723 \pm 0.0012}$ |

Table 5: Performance on Gene Expression CAGE Prediction with Standard Deviation for Cell Type GM12878.

|  | MSE ↓ | MAE ↓ | Pearson ↑ |
|---|---|---|---|
| Enformer | $0.2889 \pm 0.0009$ | $0.4185 \pm 0.0013$ | $0.8327 \pm 0.0025$ |
| HyenaDNA | $0.2217 \pm 0.0018$ | $0.3562 \pm 0.0012$ | $0.8729 \pm 0.0010$ |
| Mamba | $0.2145 \pm 0.0021$ | $0.3446 \pm 0.0022$ | $0.8788 \pm 0.0011$ |
| Caduceus | $0.2124 \pm 0.0037$ | $0.3436 \pm 0.0031$ | $0.8819 \pm 0.0009$ |
| Caduceus w/ signals | $\underline{0.1942 \pm 0.0058}$ | $0.3269 \pm 0.0048$ | $\underline{0.8928 \pm 0.0017}$ |
| EPInformer | $0.1975 \pm 0.0031$ | $\underline{0.3246 \pm 0.0025}$ | $0.8907 \pm 0.0011$ |
| MACS3 | $0.2340 \pm 0.0028$ | $0.3654 \pm 0.0017$ | $0.8634 \pm 0.0020$ |
| Seq2Exp-hard | $0.1890 \pm 0.0045$ | $0.3199 \pm 0.0040$ | $0.8916 \pm 0.0027$ |
| Seq2Exp-soft | $\mathbf{0.1873 \pm 0.0044}$ | $\mathbf{0.3137 \pm 0.0028}$ | $\mathbf{0.8951 \pm 0.0038}$ |

- The log-transformed z-score of the 3' UTR (untranslated region) length.
- The GC content of the 5' UTR, expressed as the proportion of G and C bases.
- The GC content of the CDS.
- The GC content of the 3' UTR.
- The log-transformed z-score of the total intron length for a gene.
- The exon density within the open reading frame (ORF), reflecting the number of exon junctions per unit length of the ORF.

## A.4 EXPERIMENT SETUP

Here we present some hyperparameters values and their search space in Table 3.

## A.5 EXPERIMENT RESULTS

Based on Table 1 and Table 2, here we present the experimental results with standard deviation in Table 4 and Table 5. The results are from five runs of different random seeds: {2,22,222,2222,22222}.

