# OpenReview forum: "Learning to Discover Regulatory Elements for Gene Expression Prediction"
_ICLR.cc/2025/Conference — ICLR 2025 Oral_

### Official Review · Reviewer_XARj · 2024-10-23

**Soundness:** 3
**Presentation:** 2
**Contribution:** 3
**Rating:** 6
**Confidence:** 3

**Summary:**

This paper deals with integrating genomic and epigenomic data to extract relevant regulatory DNA regions to predict gene expression, through the learning of a mask on DNA base pairs. The mask is then applied on the data, base-pair-position wise, and the masked data is used to train the predictor for gene expression. The paper describes the corresponding model in detail, and tests it alongside several baselines on genomic and epigenomic data.

**Strengths:**

- Quality: The submission seems technically correct and experimentally rigorous (however, I did not check the appendix in detail). The experimental results are convincing and appropriately compared to the numerous baselines.
- Clarity: It is well-grounded in the literature. The paper is well written and the contribution is clear.
- Originality: The paper proposes a novel decomposition of regulatory regions into three groups, that have different causal relationships to genomic and epigenomic data.

**Weaknesses:**

- Quality: The submission is not reproducible, as no code is provided (nor is the availability of the code discussed in the main paper nor in Appendix). There is no mention of hyperparameter tuning or value selection, especially for $\alpha_3$ and $\beta_3$, which parametrize the distribution of the soft mask on the regulatory regions. Performance is important for these methods, but I think runtimes across baselines and Seq2Exp should also be reported for practical purposes.
- Clarity: Some parts are not clear, for instance, the inclusion of additional data in the experiments, or the use of Hi-C data (see questions).
- Significance: The assumption of independence of selection of regulatory regions across base pair positions at fixed (epi)genomic data (assumption written at the bottom of page 4) might be a bit too strong for the real-life data (for instance, for CpG islands -although, no methylation peaks are considered in these papers). I understand that it simplifies the computations and allows for a tractable model training, but perhaps it should be discussed further in the paper.

The non-reproducibility and the unclear parts are the main reasons why I rate this paper 6 instead of 7-8.

**Questions:**

- What values were used for $\alpha_3$, $\beta_3$ in the experiments?
- Hi-C data usually gives access to a contact matrix between regions of length at least 5,000 base pairs. The appendix mentions that the paper “calculat[es] the frequency of contacts between a specific region (TSS) and all other regions, generating a Hi-C frequency distribution across the genome.” (page 15) What is exactly concatenated to the remainder of epi/genomic data? A matrix of size #base pairs x #regions that contains the frequency of contacts between the region to which a base pair belongs and all other regions? Doesn’t the number of regions change across chromosomes? Or is it a fixed number in your implementation?
- At page 8, “we incorporate additional features such as mRNA half-life and promoter activity from previous studies”. Where are those features added? To train the gene expression predictor $g_\phi$? Are they also added to all other baselines, and if so, at which point in their model of gene expression?
- Why don’t you use a neural network for also learning the parameters of the beta distribution for epigenomics ($\alpha_2$, $\beta_2$)?

---

> ### Author Response · Authors · 2024-11-22
> **Reply to Reviewer XARj**
>
> Thanks for your valuable comments, and we understand your concern about reproduction. Therefore, we provide our implementation in the following link for the reproduction: https://anonymous.4open.science/r/Seq2Exp-1E7E/README.md
> The datasets are currently too large to be hosted on an anonymous GitHub repository or provided as a zip file. Therefore, we only release our code at rebuttal. And we promise that the datasets and codes will be made publicly available after the paper is open.
> In addition, we provide our replies as follows.
>
> W1 Quality & Q1:
> * W1 Quality: We provide our implementation in the link above.
> * W1 Quality & Q1: For the hyperparameters, we summarize them in the Appendix A.4 in our revised paper. Specifically, we set α3​ and β3​ to be 10 and 90, respectively. In practice, rather than directly assigning these values, we first set the mean of the beta distribution to 0.1, for example, α=1 and β=9. We then scale these parameters by a constant factor of 10, resulting in α=10 and β=90.
> * W1 Quality: For the running time, we report the running time for one iteration of all the methods in the table below. For EPInformer, it uses part of the sequence and dilated CNN which can conduct an efficient training, and they only focus on predefined regions. And token-level sequence models like Enformer and Caduceus need longer time to encode the full sequence. Seq2Exp needs longer training time, since our framework needs to learn the positions of regulatory elements, i.e., a generator and predictor. Therefore, the training time is doubled to discover the relevant sub-sequences for the prediction.
>
> | Enformer          | Caduceus            | EPInformer            | Seq2Exp       |
> |--|--|--|--|
> | ~45min     | ~50min | ~2min | ~100min |
>
> W2 Clarity & Q2 & Q3. Thanks for pointing it out.
> * W2 Clarity & Q3: We have added a complete and precise description about these mRNA Half-life features and promoter activity in the Appendix A.3 highlighted as red to make it clear. These features, previously used in studies (Lin et al., 2024), are included in our implementation and baselines to ensure a fair comparison.  Specifically, these features are incorporated into the final output linear layer of the gene expression predictor $g_{\phi}$​, concatenated with the hidden features as input of the final layer.
> * W2 Clarity & Q2: Hi-C data is inherently represented as a matrix with a resolution of 5000 base pairs. Here, we specifically focus on the TSS, meaning we are only interested in the links between the bin containing the TSS and other bins. Although the resolution is 5000 bp, we still obtain the contact frequencies between each individual base pair and the TSS base pair. However, the contact values for a specific base pair will be identical to those of the surrounding base pairs within the same 5000 bp bin. Here we follow the  implementations of ABC model [1] and EPInformer [2].
>
> W3 Significance: Thanks for providing your concerns.
>
> Regarding the independence of mask selection between base pairs, we assume independence to facilitate the derivation of the information bottleneck and KL loss in the current framework. However, exploring "lumpy masks," which consider dependencies across base pairs, could be an interesting direction for future work. This approach has the potential to enhance the discovery of regulatory elements by capturing more complex patterns and interactions.
>
> If you ask about the Assumption 1: DNA sequences and signals are conditional independent to each other given the ground truth regulatory elements, this assumption could also benefit from further discussion. If the ground truth positions of CpG islands are known, the observed methylation peaks can be understood as a result of these positions. Moreover, DNA sequences can be viewed as a combination of regulatory and non-regulatory elements, leading to a scenario where signal measurements are independent of the DNA sequences themselves.
>
>
>
>
>
> Q4. Learning α2​ and β2​ could be an interesting direction for future work. For simplicity in our current approach, we set α2​ equal to the signal value and β2​ as a constant to incorporate the signal information. We also experimented with making β2​ learnable parameters but did not observe much differences. Future work could explore more effective ways to leverage the signal values.
>
> [1] Activity-by-contact model of enhancer–promoter regulation from thousands of crispr perturbations
> [2] EPInformer: a scalable deep learning framework for gene expression prediction by integrating promoter-enhancer sequences with multimodal epigenomic data

---

> > ### Comment · Reviewer_XARj · 2024-11-22
> >
> > Thanks for addressing my concerns. The runtime for Seq2Exp might be a bit impractical when compared to the gain in performance from EPInformer. I will keep my score to 6, but would perhaps have chosen 7 if available.

---

> ### Author Response · Authors · 2024-11-25
> **Reply to Reviewer XARj**
>
> Thank you for your response and valuable comments. We understand your concern about the training time, which is primarily due to our framework using the long DNA sequence. DNA language models, such as Caduceus and HyenaDNA, have demonstrated their effectiveness across various downstream tasks while maintaining reasonable training time with long DNA sequence. However, the integration of these models with epigenomic signals remains underexplored. Incorporating such signals could provide valuable insights and pave the way for powerful models capable of handling multiple cell types in future research. Therefore, while our method requires higher computational costs due to its use of long DNA sequence, we believe it is crucial to develop a learnable way to consider these epigenomic signals modeling to discover the regulatory elements. Moreover, developing more efficient and trainable methods to achieve this remains an important and promising direction for further exploration. Once again, thank you for your response and your willingness to support us with a score of 7. We truly appreciate it.

---

> > ### Comment · Reviewer_XARj · 2024-11-25
> >
> > Thanks for your reply, which brings an excellent argument in favor of the methods developed in this submission.

---

### Official Review · Reviewer_AbGL · 2024-11-01

**Soundness:** 4
**Presentation:** 3
**Contribution:** 3
**Rating:** 8
**Confidence:** 3

**Summary:**

This paper describes a model, Seq2Exp, that predicts gene expression values from two inputs: DNA sequence and a corresponding matrix of epigenomic measurements.   A key component of the model is a binary masking strategy that aims to automatically identify relevant regulatory elements in the DNA sequence.  The model uses an information bottleneck approach to try to ensure accurate extraction of relevant sequence signals.  The method is compared to a variety of state-of-the-art methods and yields better performance according to a variety of measures.

**Strengths:**

The main idea is well articulated and sensible.

The paper is very clearly written.

The related work section does a reasonable job of covering the recent, relevant literature.

The model is well formulated and modeling choices are well motivated in the text.

The empirical results are strong.

**Weaknesses:**

I think the sentence "Gene expression prediction is one of the fundamental tasks in bioinformatics." should be supported with a citation or two from some of the seminal works in this area, e.g., Nir Friedman's 2000 paper on predicting gene expression from promoter sequences.

The task description should be expanded.  It's not clear, from the description, how the inputs are registered to the output value Y.  Is the idea that the length-L input window is centered on the TSS?  To my mind, it would be clearer to represent Y as a vector over genes, and then include some formalism to represent the indexing of genes relative to X_{seq} and X_{sig}.

Line 420 ("Furthermore, we incorporate additional features such as mRNA half-life and promoter activity ...") is problematic because it sounds like this is an incomplete list.  For reproducibility, all inputs should be precisely described.  These features are not mentioned at all in Appendix A.3.

I don't see any point in using both R and R^2 as two distinct performance measures.  One is just the square of the other.

Minor point, but I think you should mention in the description of the baseline methods the size of the field of view of each one.

**Questions:**

A priori, I would expect the true binary mask to be "lumpy," i.e., groups of nearby base pairs should tend to be either relevant or irrelevant, reflecting the fact that, e.g., TFs tend to bind to regions rather than single base pairs.  How does your model handle this?

Section 5.1.2 does not make clear whether the competing methods were trained using your cross-chromosome train/test splits.  Is that the case?  I think you should clarify this point in the paper and also discuss how it may impact your results.  If you trained from more or a different type of data than your competitors, the comparison might be unfair.  OTOH,  if your competitors trained on data from your test chromosomes, then it might be unfair in the other direction.

I like the idea of Section 5.3, but I had trouble understanding exactly what was done here. MACS3 is a peak caller, so what is the model that predicts expression in this setting?

---

> ### Author Response · Authors · 2024-11-22
> **Reply to Reviewer AbGL**
>
> Thank you for your valuable comments. We have provided the replies as follows.
>
> W1. Thanks for your suggestion. We have added this citation to support this sentence.
>
> W2. We have added the detailed implementation about how to select the window around the target gene in Section 2.1 task description as follows.
>
> Specifically, in our implementation, each example contains one target gene. We first identify the transcription start site (TSS) of the target gene, then select input sequences $X_{seq}$ and $X_{sig}$ consist of $L = 200,000$ base pairs, centered on the TSS. Then, the entire sequences provide sufficient contextual information for accurate prediction of the target gene expression value $Y$.
>
>
> W3. Thanks for pointing it out. We have added a complete and precise description about these mRNA Half-life features in the Appendix A.3 highlighted as red to make the list complete.
>
> W4. Thanks for pointing it out. We removed the $R^2$ metric and retained the Pearson correlation in our experiment in our revised paper shown in Table 1 and Table 2.
>
> W5. Description about the size of the field for each method is summarized as follows.
> The size of the field of view of Enformer, HyenaDNA, Mamba, Caduceus are the full sequence. And the size of the field of view for EPInformer is the extracted potential enhancer candidates based on DNase-seq measurement following ABC model [5]. Our proposed Seq2Exp also has the view of full sequence, since the generator will take the full sequence to extract the relevant sub-sequences for the prediction.
>
>
> Q1.
> Thanks for posting this question. Currently, our model uses single base pair masks instead of continuous lumpy masks. Exploring lumpy masks could be an interesting direction for future work, as it can provide a valuable prior and enhance the discovery of regulatory elements.
>
>
> Q2. Thanks for posting this question. We would like to clarify that the proposed Seq2Exp and other baselines are all under the same train/val/test dataset to make a fair comparison.
> Specifically, chromosomes 3 and 21 are used as the validation set, and chromosomes 22 and X are reserved for the test set. And all other chromosomes are used in training. You can find this description in Section 5.1.4.
>
>
> Q3. We would like to clarify that we train a predictor model based on the mask obtained from MACS3 and compare its performance to that of a predictor model trained on regions discovered by our framework.
> The motivation of this experiment is to determine which method—MACS3 or our framework—can discover regions more closely related to target gene expression.
> Statistical peak-calling methods, such as MACS3, can be considered a straightforward approach for measuring these elements. This methodology has been utilized by models like the ABC model [5] and EPInformer [4].
> Our final prediction results demonstrate that the regions found by our framework enable the predictor model to achieve better performance shown in Table 2, indicating that our framework discovers better regions than those identified by statistical methods.
>
>
> [1] Effective gene expression prediction from sequence by integrating long-range interactions
>
> [2] Mamba: Linear-Time Sequence Modeling with Selective State Spaces
>
> [3] HyenaDNA: Long-Range Genomic Sequence Modeling at Single Nucleotide Resolution
>
> [4] EPInformer: a scalable deep learning framework for gene expression prediction by integrating promoter-enhancer sequences with multimodal epigenomic data
>
> [5] Activity-by-contact model of enhancer–promoter regulation from thousands of crispr perturbations

---

> > ### Comment · Reviewer_AbGL · 2024-11-25
> >
> > Thank you for the detailed responses. I think the revised version of the manuscript is much better. Please add some information from your response to W5 above into the manuscript.

---

> ### Author Response · Authors · 2024-11-26
> **Reply to the Reviewer AbGL**
>
> Thanks for your suggestions. We have included the information about the response to W5 in Section 5.1.2 of the new revised paper. Once again, we sincerely appreciate your support.

---

### Official Review · Reviewer_KQDw · 2024-11-04

**Soundness:** 4
**Presentation:** 3
**Contribution:** 3
**Rating:** 8
**Confidence:** 4

**Summary:**

The paper is aimed at predicting gene expression from DNA sequence. In addition to the sequence itself, it also incorporates epigenetic information at each based. The model consists of two MAMBA-based Caduceus networks: the first network is trained to estimate a sparse mask over the bases to select regulatory regions, and the second one uses the selected DNA regions to predict the expression of the gene.

**Strengths:**

The paper showcases a successful application of the recently-proposed Selective State Space models (specifically, MAMBA-based Caduceus) on relatively long sequences (200k).

The model goes beyond most current approaches by combining sequence with epigenetic signaling in a single, jointly optimized architecture. The architecture is sound, and uses information bottleneck principle to promote sparsity of the mask for selecting sequence regions, and then a straight-through estimator to allow differentiability through the hard mask being produced by the first network.

Experimental evaluation shows that the proposed architecture leads to improved prediction, in comparison with Transformer- and State-space-based models.

**Weaknesses:**

While the manuscript is mostly well written, it could use a more detailed explanation in several places, specifically related to:
- gene expression Y: it is often just described as "target variable Y", or "$Y \in R$. From 5.1.4 we read that ultimately the model is applied to a 200kbp region around a specific target gene, and one can understand that Y is the expression of that gene; it would be helpful to provide this information clearly earlier in the paper.
- model architecture: it also deferred to 5.1.4, and only described in one sentence (Caduceus-Ph). One would expect it to be in 4.1. entitled "Model architecture". Also, key details are missing: what are the specific hyperparameters (#layers, hidden dimension, etc.)? Is this a network trained de-novo for the expression task, or pre-trained (Caduceus has pretrained models of up to 131kbp, this one is 200kbp, if it was pre-trained by the authors, more details should be provided).

**Questions:**

Details about the network training (and pre-training, if used), including architectural hyperparameters, as well as training procedure details (# steps, etc).

---

> ### Author Response · Authors · 2024-11-22
> **Reply to Reviewer KQDw**
>
> Thank you for your valuable comments. We have revised our paper to make it more clear for the readers.
>
> W1. We have changed the description from ‘target Y’ or ‘$\mathbf{Y \in R}$’ to target gene expression $Y$ or add some additional sentences to further explain these symbols. We highlight these modifications in red in the revised paper. Moreover, we add the description about the model input and target gene expression in Section 2.1 task description to provide these details at the beginning of our paper to make it more clear.
>
> W2. Thanks for pointing it out. We have reorganized the content to section 5.1.4, providing additional details as follows:
> Both generator $p_\theta$ and predictor $q_\phi$ are based on Caduceus architecture [1], an advanced long-sequence model considering the bi-direction and RC-equivariance for DNA. Specifically, we train for 50,000 steps on a 4-layer Caduceus architecture from scratch with a hidden dimension of 128, and more hyperparameters can be found in the Appendix A.4.
>
> Q1. Thanks for pointing it out. We have added these hyperparameters in Appendix A.4.
>
> [1] Caduceus: Bi-Directional Equivariant Long-Range DNA Sequence Modeling

---

> > ### Comment · Reviewer_KQDw · 2024-11-25
> >
> > Thank you for the additional information.

---

### Official Review · Reviewer_hbLE · 2024-11-09

**Soundness:** 3
**Presentation:** 3
**Contribution:** 3
**Rating:** 8
**Confidence:** 4

**Summary:**

In this paper the authors have highlighted the importance of gene expression as the fundamental process in biology, and have introduced a framework for predicting gene expressions by leveraging causal relationships between DNA sequences, epigenomic signals and regulatory elements or motifs. They have provided viability of this approach over two cell types and also state that adding the "extra info" in form of epigenomic signals is not always useful and may act as noise in some prediction tasks. To agree on generalization of this approach it has to be tested on more cell types and diverse epigenomic data.

**Strengths:**

When I compare this paper with other benchmarks (except EPInformer) in the field and for the same problem definition - I would rate this paper as high in terms of originality & significance. Yes, this work shows that gene expression prediction is not a siloed process, and that there are causal relationships between DNA sequences, epigenomic data, regulatory elements, and surrounding causal / non-causal parts of the input data. Also, deep learning algorithms can learn from these relationships and perform better gene expression predictions.

**Weaknesses:**

This follows from my comments on the strengths of the paper. After reading the EPInformer paper (released in early 2024), my rating reduces. Was this idea a new approach? I respectfully decline. EPInformer highlights the same intent, experimental approach and even comes very close in performance benchmarks & standard metrics. Not sure if this paper has added anything new for the ICLR community? To me as a reader, it has only reinforced the claims of EPInformer. I would have been happier if they can show viability of this approach over multiple cell types. But, data availability may be a limitation here.

Update: Thanks to the authors for reviewing my feedback and responding with relevant updates in the paper and their explanation. My concerns are settled. I would like to update my rating for this paper accordingly.

**Questions:**

Table 1 shows good comparison with previously available benchmark models and datasets. However, looking at EPInformer row, my question is did the authors run the EPInformer code or model over the CAGE prediction data? Because I only saw Pearson metric in the EPInformer paper. How were they able to get MSE, MAE, R2 numbers for comparison?

In the same table, GM12878 CAGE prediction numbers between EPInformer and Seq2Exp are exactly the same at two decimal places (not a significant improvement). This also indicates that the approach presented in the two papers is very similar. Neither one is better than the other. Originality in terms of problem definition and solution is also similar.

Also, some numbers do not match between the EPInformer row in this paper and the original EPInformer paper. For example, in the original EPInformer paper, figure 2C, Pearson coefficient for K-562 on CAGE-seq hold out test set is 0.848. But, the same info in Seq2Exp paper, table 1, says is 0.8517. How come?

Update: Questions / Concerns were settled by reply from the authors. Thanks!

---

> ### Author Response · Authors · 2024-11-22
> **Reply to Reviewer hbLE**
>
> Thanks for your comments, and we provide more information about the weaknesses and questions.
>
> W1. Thank you for sharing your concerns regarding the contributions of our proposed method compared to EPInformer.
> EPInformer builds upon the annotated enhancer positions identified by the Activity-By-Contact (ABC) model [1] and then leverages deep learning to predict gene expression. Its primary contribution lies in effectively utilizing these pre-annotated enhancer regions. In the genomics field, signal peaks derived from measurements like DNase-seq or H3K27ac often serve as indicators of enhancer positions. Essentially, EPInformer’s methodology relies heavily on these signal measurements, denoted as Rm​ in Figure 1 of our paper.
> However, EPInformer does not account for Rg​, the possibility that measured enhancer peaks may not interact with the gene of interest. This limitation motivates our approach, which integrates DNA sequence information and utilizes deep learning rather than solely relying on statistical measurements to discover regulatory elements.
> One of the core contributions of Seq2Exp lies in its ability to discover regulatory elements (including enhancers) using the principle of information bottleneck. To the best of our knowledge, we are the first to employ deep learning for this purpose, whereas prior studies focus on the results of signal measurements. As demonstrated by our updated experimental results, Seq2Exp outperforms EPInformer in performance.
> Furthermore, we implement a new version: Seq2Exp-soft, which learns the soft importance values instead of the hard mask within our framework, indicating an improved performance.
>
> Moreover, we include an additional cell type, H1, and apply both our method and the baseline EPInformer for evaluation below. Additionally, we ran experiments with five different random seeds for K562 and GM12878 as part of the rebuttal. Due to time constraints, we provide the results of a single run on H1 using the soft version in the rebuttal. Results for five random seeds, the hard version, and all other baselines on H1 will be reported when the training is finished.  You can find more details about experiments in our responses to Q1, Q2, and Q3. We wish to convey that the performance improvement observed across most experimental settings cannot be ignored.
>
> Q1 & Q3. Thanks for posting your concerns, we wish to clarify that we use the codes of EPInformer to re-run the experiments, with the totally same setting as the original paper. In this way, we get the MSE, MAE, Pearson coefficient. To make the results more trustable, we additionally run the experiments with five different random seeds on K562 and GM12878 for the rebuttal and provide the results as below.  Note that we are running multiple runs results on other baselines and hard versions, and we will report these in our paper when they are finished. In addition, we also provide our implementation in this link for reproduction: https://anonymous.4open.science/r/Seq2Exp-1E7E/README.md
>
> Q2. We understand your concerns.
> We wish to convey that in most metrics and cell types, our proposed methods have surpassed the baselines methods greater than the std indicating a valid improvement. Specifically, on GM12878, the error reduction of Sep2Exp-hard on MSE and MAE is a little larger than or equal to the std of EPInformer experiments, indicating a valid improvement.  Furthermore, to address your concern, we improve the performances of our framework with a soft mask version. Specifically, on GM12878, Seq2Exp-soft achieves an error reduction of more than 0.01 in both MAE and MSE compared to EPInformer. **Note that 0.01 is about 5% and 3% improvement on MAE and MSE for GM12878, respectively.** Furthermore, when the top 10% is selected from a soft mask for retraining, MAE can also achieve an error reduction of about 0.01.
> Moreover, in the experiment of K562, the improvement of both soft version and hard version is clear. We also provide the updated results as below and in our revised paper.
>
> [1] Activity-by-contact model of enhancer–promoter regulation from thousands of crispr perturbations
>
> ### H1
> | Model| MSE ↓ | MAE ↓ | Pearson ↑|
> |--|--|--|--|
> | EPInformer  | 0.2911 | 0.4005 | 0.6340   |
> | Seq2Exp-soft   | 0.2724 | 0.3913 | 0.6684 |
>
> ### K562
> | Model   | MSE ↓  | MAE ↓  | Pearson ↑   |
> |--|--|--|--|
> | EPInformer     | 0.2140 ± 0.0042 | 0.3291 ± 0.0031 | 0.8473 ± 0.0017 |
> | Seq2Exp-hard   | 0.1951  | 0.3150   | 0.8623 |
> | Seq2Exp-soft   | 0.1856 ± 0.0032 | 0.3054 ± 0.0024 | 0.8723 ± 0.0012 |
> | Seq2Exp-retrain| 0.2001 ± 0.0058 | 0.3181 ± 0.0056 | 0.8612 ± 0.0026 |
>
> ### GM12878
> | Model   | MSE ↓    | MAE ↓  | Pearson ↑  |
> |--|--|--|--|
> | EPInformer     | 0.1975 ± 0.0031 | 0.3246 ± 0.0025 | 0.8907 ± 0.0011 |
> | Seq2Exp-hard   | 0.1900  | 0.3221    | 0.8942  |
> | Seq2Exp-soft   | 0.1873 ± 0.0044 | 0.3137 ± 0.0028 | 0.8951 ± 0.0038 |
> | Seq2Exp-retrain| 0.1880 ± 0.0026 | 0.3172 ± 0.0018 | 0.8960 ± 0.0009 |

---

> ### Author Response · Authors · 2024-11-25
> **Reply to Reviewer hbLE**
>
> Dear reviewer hbLE,
>
> As the discussion period is approaching its deadline. We would appreciate knowing if our responses have addressed your concerns. If you have any remaining questions or concerns, please don't hesitate to share them with us, we will be happy to respond! If we have sufficiently alleviated your concerns, we hope you'll consider raising your score. Thank you for your time and consideration!

---

> ### Comment · Reviewer_hbLE · 2024-11-26
> **Update to previous review from Reject to Accept.**
>
> This is definitely good. Thanks for replying to my questions. I've reviewed the updated draft and author responses, and as a result have updated my rating to this paper. As a reader, I now understand "what new" approach they bring to the table for prediction gene regulation. Please open source your code and likely provide a library for ease of use.

---

> > ### Author Response · Authors · 2024-11-28
> > **Reply to Reviewer hbLE**
> >
> > Thank you for your valuable suggestions. We will release the code and ensure that all datasets and model checkpoints are available once the paper is open. Your support is greatly appreciated!

---

### Meta-Review · Area_Chair_ka1N · 2024-12-18

**Metareview:**

This paper describes a model, Seq2Exp, that predicts gene expression values from sequence and epigenetic data. The method claims state-of-the-art performance compared to gene expression prediction baselines.

The reviewers identified numerous strengths of the paper. The reviewers found the binary masking strategy interesting and well-motivated. The empirical results were strong. And the clarity was good.

Some weaknesses were identified in the review process particularly around novelty compared to EPInformer. But these weaknesses were addressed and clarified by the authors and revised in the submission.

The paper combines a relatively novel idea to an important problem in computational biology. The claims rest comfortably upon the evidence provided and the approach seems to achieve state-of-the-art performance. For these reasons I recommend acceptance.

**Additional Comments On Reviewer Discussion:**

There was a productive discussion between the authors are Reviewer hbLE that are much appreciated because it clarified the novel contributions of the paper. Reviewer AbGL pointed out some important references to contextualize the contribution. Overall, the discussion significantly improved the paper and the authors were responsive to the comments.

---

### Decision · Program_Chairs · 2025-01-22

Accept (Oral)